# Cellulose Amphiphilic Materials: Chemistry, Process and Applications

**DOI:** 10.3390/pharmaceutics14020386

**Published:** 2022-02-10

**Authors:** Simona Zuppolini, Ahmed Salama, Iriczalli Cruz-Maya, Vincenzo Guarino, Anna Borriello

**Affiliations:** 1Institute of Polymers, Composites and Biomaterials, National Research Council of Italy, Mostra d’Oltremare, Pad. 20, 80125 Naples, Italy; simona.zuppolini@cnr.it (S.Z.); cdiriczalli@gmail.com (I.C.-M.); borriell@unina.it (A.B.); 2Cellulose and Paper Department, National Research Center, 33 El-Bohouth St., Dokki, Giza P.O. Box 12622, Egypt; ahmed_nigm87@yahoo.com

**Keywords:** biopolymers, polysaccharides, drug delivery, antibacterial inhibition

## Abstract

In the last decade, amphiphilic cellulose (AC) is emerging as attractive biomaterial for different therapeutic use, due to its unique chemical and physical properties. Using it as alternative to synthetic polymers, AC opens up new avenues to prepare new bio-sustainable materials with low impact in the cellular environment. Herein, most recent methods to synthesize and processing AC materials from different sources—i.e., cellulose nanofibers, bacterial cellulose, cellulose derivatives—will be discussed. By an accurate optimization of morphology and surface chemistry, it is possible to develop innovative amphiphilic platforms, promising for a wide range of biomedical applications, from drug delivery to molecular/particle adsorption.

## 1. Introduction

Cellulose is the most widespread natural raw polymer material and the most abundant polysaccharide in plants and derivatives (trees, cotton, straw) [1]. Cellulose acts as scaffolding and reinforcement material in plant cell walls due to its tough, fibrous, and water-insoluble properties and produced through photosynthesis [2]. Its chemical structure is a highly hydrophilic homopolymer consisting of a β(1-4)-glucopyranoside as a structurally repetitive unit (Figure 1), which is a anhydroglucose dimer. The number of linked anhydroglucose (C_6_H_10_O_5_) unities usually ranges from 10,000 to 15,000 depending on the cellulose source material as well as the extraction conditions for purification [3,4]. The fully equatorial conformation of β-linked glucopyranose residues stabilizes the chair structure, minimizing its flexibility. The anhydroglucose unit bears three hydroxyl groups that form strong hydrogen bonds.

Although hydrophilic, cellulose in its native form is not soluble in water or conventional organic solvents due to numerous strong intra- and inter-molecular hydrogen bonds between individual chains (Figure 2) [5]. The regular structure of cellulose, as indicated by X-ray diffraction patterns, is crystalline. In particular, X-ray diffraction patterns and solid-state ^13^C-NMR discovered cellulose conformations that explain the detailed crystalline structure and the basis for transformation in the different allomorphs [6]. This complex H-bond network, in combination with a lot of van der Waals interactions between residues on contiguous sheets (inter sheet), permits the formation of a very packed molecular arrangement and hence facilitates the establishment of three-dimensional structures with a high degree of crystallinity (in the range of 40–60%). Cellulose displays different polymorphisms with the possibility of conversion from one form to another [7]. Cellulose I, crystalline cellulose, is thermodynamically metastable and can be converted to either cellulose II or III. Solid state ^13^C-NMR studies revealed that cellulose I can crystallize in two different phases: cellulose Iα (monoclinic unit cell) and Iβ (triclinic symmetry). Cellulose II, the most stable form, can be obtained by regeneration or via alkaline treatments, and cellulose III can be formed from cellulose I or II after liquid ammonia handling. Cellulose II contains two chains of cellulose located antiparallel to the monoclinic cell. Moreover, cellulose IV can be formed after subsequent thermal treatments [8].

In (1,4)-linked polysaccharides, anhydroglucose units present in a chair conformation where hydrophilic hydroxyl groups are in equatorial position and the hydrophobic C–H bonds are in an axial position. α-(1,4)-linked polysaccharides form a cavity rich in C–H bonds that is suitable for encapsulating hydrophobic chemicals, including for delivery in living organisms [9].

Cellulose molecules accumulate to form cellulose microfibrils, which are characterized by the presence of either highly ordered (crystalline) or less-ordered (amorphous) regions (Figure 3). Cellulose macrofibers are composed of microfibrils, which are in turn formed from nanofibrils that have a crystalline part and an amorphous part in a row. However, celluloses from different sources may occur in different packing as dictated by biosynthesis conditions. Cellulose displays distinctive properties, such as a low density (1.5–1.6 g·cm^−3^), high tensile strength (290–600 MPa for crystalline cellulose), and a high Young’s modulus (100–140 GPa for the crystalline region) [10]. However, poor solubility and low functionality hamper its application in different fields. The intermolecular and intramolecular hydrogen bonds render the cellulose fibers insoluble in most solvents. Various chemical modifications such as etherification, esterification, cross-linking, or graft-copolymerization reactions were recently applied to prepare functionalized cellulose [11].

## 2. Different Cellulose Forms

### 2.1. Cellulose Nanocrystals (CNCs)

Individual cellulose molecules are brought together into larger units known as microfibrils, which are packed into larger units called micro-fibrillated cellulose. The diameter of these microfibrils is around 20 to 50 nm. The microfibrils are formed during the biosynthesis of cellulose and are several micrometers in length. The ordered regions are cellulose chain packages that resemble nanocrystalline rods.

These nanocrystalline parts contain highly ordered (crystalline) domains which can be separated and extracted from different cellulose sources through acid hydrolysis using sulfuric, hydrochloric, or phosphoric acid. Acid hydrolysis facilitates the cleaving of accessible glycosidic linkages within the cellulose microfibrils resulting in the characteristic rod-shaped nanocrystals [12]. CNCs have unique features such as being lightweight and stiff and having a high mechanical reinforcing capability and renewability compared to neat cellulose. CNCs exist in different morphologies in terms of the length, width, aspect ratio (length to diameter about 1:10), and shape, which can vary from rod-like to spherical, depending on the cellulose origin and hydrolysis conditions. Sulfuric acid hydrolysis enhances the removal of amorphous regions embedded within cellulose microfibrils, leading to highly crystalline particles with charged surfaces. The presence of sulfate half-ester groups at the CNC surface during hydrolysis gives them electrostatic stabilization upon suspension in water and with exciting self-organization [13]. These charges increase hydrophilicity and electrostatic repulsions, and the negatively charged surfaces permits the formation of stable colloidal suspensions [14].

Cellulose nanocrystals (CNCs) are usually considered to be isotropically polar nanoparticles with amphiphilic behavior in suspension. The Hansen solubility parameters of wood-based H_2_SO_4_-hydrolyzed CNCs were measured from sedimentation tests in a wide set of 59 solvents and binary mixtures [15]. It was reported that the polar sphere is thought to correspond to the (110) surfaces of cellulose I nanocrystals, while the smaller non-polar sphere is coherent with the exposure of (200) surfaces. The Hansen solubility parameters graph provides new insights into the amphiphilic nature of CNCs and the mapping of their chemical affinity for solvents and polymer matrices [8,16].

Moreover, CNCs have rich functional moieties, such as aldehyde and hydroxyl groups, which offer a variety of modifications to control their physical or chemical properties. The presence of an aldehyde groups at the end of a CNC permits the selective modification of reactive reducing ends to prepare amphiphilic CNCs with special features. Otherwise, some relevant limitations in the use of CNCs for particle entrapment concern the stabilization of the polymers from oil-water emulsions. Different modifications were carried out to increase the hydrophobicity of CNCs. For example, end-group modification by the introduction of hydrophobic polystyrene chains was carried out to increase the viability of modified CNCs in emulsifying toluene and hexadecane compared to unmodified CNCs [17]. Moreover, cellulose nanocrystals were modified to produce amphiphilic nanoparticles that improve the stability of emulsions. This kind of nanoparticle showed biocompatible and environmentally friendly features and has promising applications in the food, drug delivery, cosmetics and petrochemical industries [17].

Recently, nanocellulose has been applied in drug delivery applications (Table 1). However, despite its considerable merits, the inherent hydrophilic structure of nanocellulose limits its applications for the delivery of hydrophobic drugs. Grafting hydrophobic polymers onto the surface of CNCs could provide the polymers with amphiphilic features and thus form amphiphilic copolymers [18].

Cellulose nanocrystals, with many hydroxyl and sulfate groups on the surface, can stabilize metal nanoparticles effectively. Moreover, CNCs derived from sulfuric acid hydrolysis are super-hydrophilic and show great dispersibility in water because of the abundant hydroxyl and surface sulfate ester groups, which make great supports for heterogeneous catalysts for hydrodeoxygenation reactions. In the field of amphiphilic cellulose nanocrystals, benzyl-polyethyleneimine modified cellulose nanocrystals were developed via the periodate oxidation of cellulose nanocrystals and reductive amination, which exhibited pH-responsive amphiphilicity due to the existence of hydrophilic amino- and hydrophobic benzyl groups [17].

### 2.2. Cellulose Nanofibers (CNFs)

Cellulose nanofibers (CNFs) have been used as an alternative non-toxic and bioactive material in a broad field of innovative nanostructured materials (Table 1). CNFs were prepared from natural cellulose using high mechanical shearing. Different pretreatment techniques were reported to facilitate defibrillation and increase the yield in cellulose nanofibrils such as TEMPO-mediated oxidation, enzymatic hydrolysis, carboxymethylation and mechanical refining [29]. Mechanical refining, grinding or shearing, such as high-pressure homogenization or can enhance the progressive release of constitutive CNFs. The high-pressure homogenization of cellulose fibers yields CNFs with diameters of ~20–40 nm and several micrometers in length. Moreover, the oxidation with 2, 2, 6, 6- tetramethylpiperidine-1-oxyl (TEMPO) oxidizes the primary hydroxyl groups into carboxyl groups [30]. The diameters of CNFs, produced by TEMPO-mediated oxidation after successive mechanical treatment by high-pressure homogenization, show uniform widths, ca. 3–4 nm, a high aspect ratio (greater than 100) and a high elastic modulus of 145 GPa [31]. The principle of these pretreatments depends on weakening the hydrogen bonds, adding a repulsive charge and breaking down the amorphous links between CNFs. Xylanase-pretreatment has been reported as an efficient method for preparing CNFs from cellulose pulp. This process reduces the hemicellulose content while enhancing the degree of fiber polymerization. The enzymatic pretreatment is an effective technique that facilitates the disintegration of the fibers. CNFs produced by this technique exhibited a higher average molar mass and larger aspect ratio than nanofibers achieved by acidic pretreatment only [14].

Moreover, the surface-bound charged CNFs were reported to act as dispersing and structural-forming agents for multifunctional exfoliating. Amphiphilic CNFs attract with graphite through hydrophobic interactions, whereas hydrophilic surfaces assist in the dispersion of calcium phosphates in aqueous media [30]. Moreover, the carboxylate groups assist the CNF-bound graphene dispersions in water via Coulomb repulsion [32].

### 2.3. Bacterial Cellulose

Cellulose can be produced by organisms such as vascular plants, algae, certain types of bacteria, and even some animals [33]. Bacterial cellulose has a unique sort of cellulose nanofibers with a diameter of approximately 50–100 nm. It is secreted extracellularly by specific bacteria, mainly Gluconacetobacter strains. Its unique fibrillar nanostructure endows it with excellent physical and mechanical properties such as an ultrafine reticulated structure, high porosity, high tensile strength, high crystallinity, and hydrophilicity although its chemical structure is similar to plant cellulose [7]. Bacterial cellulose nanofibers with a three-dimensional network structure (pellicle) are promising materials for various applications. Generally, nanofibrillated bacterial cellulose has low dispersibility in organic solvents. Different methods such as surface hydrophobic treatments are applied to change their properties [34]. Cellulose nanocrystals prepared by the acid hydrolysis of bacterial cellulose, are environmentally friendly, nontoxic, edible, degradable and biocompatible [35]. Bacterial cellulose nanocrystals exhibit an overall amphiphilic composition due to the high density of surface hydroxyl groups, while the hydrophobic interactions are caused by the crystalline structure and extensive hydrogen bonds in polymer chains. Therefore, their amphiphilic properties could be applied to stabilize surfactant-free Pickering emulsions. For example, the loading and continuous release of alfacalcidol for composite beads was achieved by the interfacial assembly of amphiphilic bacterial cellulose nanocrystals in oil-in-water Pickering emulsions followed by external gelation. In oil–water interfaces, the emulsifying effect of bacterial cellulose nanocrystals increases the compatibility between alginate and alfacalcidol. Moreover, in composite beads formed by external gelation, the hydrogel shells are responsible for encapsulating and releasing alfacalcidol [36].

### 2.4. Chemical Modification: Cellulose Derivates

Cellulose and its derivatives have been demonstrated to be non-toxic in both animals and humans, which makes it an ideal material for biomedicine [37]. Cellulose is a subject of essential research, and new applications have emerged from this field. Various trials have been carried out to enhance its function as a promising candidate for biomedical applications, especially drug carrying systems. Although hydrophilic, the poor solubility of cellulose is the major obstacle to its use in new applications.

Many attempts have been carried out to overcome this problem. All commercially available cellulose products are obtained by solid phase reactions, more or less swollen, state (heterogeneous reactions) because of insolubility of cellulose caused by supramolecular structure [16]. Through the use of specific solvents that break up the inter- and intramolecular hydrogen bonds [35], the influence of the supramolecular structure of cellulose on the reaction is eliminated almost completely. Of course, partially substituted soluble cellulose derivatives are also good substrates for reactions under homogeneous conditions. Various solvent systems have been developed to dissolve cellulose, such as ionic liquids, which have been applied as solvents and reaction media for cellulose reactions. The most available cellulose solutions are prepared with 1-ethyl-3-methylimidazolium acetate due to its lower viscosity and high cellulose dissolving ability. Ionic liquids have been successfully used as reaction media for homogeneous cellulose derivatizations [1]. Cellulosic materials synthesized from ionic liquid-based process yields materials that are potentially beneficial for regenerative therapies [38].

The most promising strategy to use cellulose is taking in advantage with the natural abundance of different hydroxyl groups to attempt chemical modifications. Cellulose derivatives can be synthesized not only by etherification or esterification of neat cellulose, but also oxidation, silylation and polymer grafting [10].

There are many kinds of cellulose derivatives used frequently, such as cellulose acetate (CA), methylcellulose (MC), ethylcellulose (EC), hydroxyethylcellulose (HEC), hydroxypropylcellulose (HPC), sodium carboxymethylcellulose (CMC) and others.

CMC a water-soluble cellulose ether, can be prepared from the reaction of cellulose with sodium monochloroacetic acid. CMC can be characterized by different degrees of substitutions (ranging from 0.4 to 1.3) depending on the average number of carboxymethyl groups per glucose unit. CMC degrades completely at low rates in the environment to form nontoxic intermediates [39]. Moreover, its chemical modifications can offer additional desired properties to enhance its potential applications. Other water cellulose derivatives such as ethyl cellulose [40] and hydroxy alkyl cellulose [41] have been studied for preparing amphiphilic celluloses.

Due to the low processability of neat cellulose, the synthesis of cellulose derivatives such as cellulose esters or ethers generates valuable properties, including suitable hydrophobicity/hydrophilicity, thus enabling the use of cellulose derivatives in various applications. Dong et al., demonstrated new cellulose ether-based amphiphiles, cellulose ω-carboxyalkyl derivatives, which prevented crystallization of supersaturated solutions of poorly water-soluble drugs. The authors prepared cellulose ethers with the required balance of hydrophobicity/hydrophilicity through two techniques: starting from a hydrophilic methyl cellulose or by a one-pot homogeneous reaction. A series of cellulose ethers with an adaptable degree of substitution and proper and terminal functionality were prepared. These derivatives also inhibited nucleation from supersaturated solutions of the poorly water-soluble drug and enhanced their potential use as amorphous solid dispersion [42].

## 3. Amphiphilic Cellulose

Amphiphilic celluloses (ACs) combine hydrophilic parts and hydrophobic moieties. The hydrophilic parts are responsible for hydration and swelling, while the hydrophobic moieties minimize water contact. Introducing or grafting hydrophobic segments to hydrophilic cellulose represents a promising tool for constructing amphiphilic systems. There are two main strategies for constructing an amphiphilic system: covalent conjugation and non-covalent intermolecular interaction. In covalent conjugation, cellulosic materials containing hydroxyl groups can react with a wide range of hydrophobic moieties via ester bonds to form amphiphilic polymers. The covalent chemical bonds between cellulose and hydrophobic moieties can be used in biomedical applications especially drug release. A non-covalent strategy may refer to electrostatic interactions, host–guest recognition, hydrogen bonding, or charge transfer interaction [43].

Because they have both hydrophobic and hydrophilic parts, amphiphilic cellulose copolymers can self-assemble into micelle structures when in contact with water or a selective solvent [44]. Particularly in aqueous media, micelles have a hydrophobic inner core and a hydrophilic outer shell that stabilizes the interface between the core and the aqueous surroundings, and vice versa in organic solvents. Hence, amphiphilic cellulose has recently emerged as a promising material that can self-assemble into core-shell micelles and consequently act as a host material for drugs and other active materials. Due to the excellent properties of micelles (e.g., solubilization and small size), they enhance drug bioavailability and reduce drug side effects. The size, shape and surface chemistry of the formed nanoparticles was found to have an impact on cellular uptake and the efficiency of drug delivery, especially hydrophobic drugs [45].

Various amphiphilic polymers such as carboxymethyl cellulose acetate butyrate have gained increasing importance in coating technologies and pharmaceutical research. Carboxymethyl cellulose acetate butyrate can be prepared by esterification of CMC with acetic and butyric anhydrides. The main character of carboxymethyl cellulose acetate butyrate is its dispersity in water to form stable colloidal solution. Moreover, its ability to control the release rate of highly water-soluble compounds and enhance the dissolution of poorly water-soluble drug proposes variety of applications especially in pharmaceutical and drug delivery applications [46].

### 3.1. Cellulose Graft Modifications

#### 3.1.1. Amphiphilic Cellulose Graft Copolymers

One convenient route for introducing new chemical and physical properties to cellulose is graft modification. In contrast to block copolymers, graft copolymers have special architectures that contain a main chain or backbone and many side-chains, which could cause a unimolecular association and result in the unimolecular micelles (Figure 4). In the majority of cases, graft copolymers in selective solvents tend to form spherical micelles by stable thermodynamic interactions among side chains and the backbone [47] until to reach the critical micellar concentration (CMC) thanks to the presence of surfactants [48].

In the case of grafted copolymers, however, additional parameters, mainly graft density and graft chain length, should be taken into account to correlate the molecular and micellar characteristics [19].

Amphiphilic graft copolymers are generically composed of a hydrophilic main chain and hydrophobic branches which are spaced comparatively closely along the main chain. After appropriate chemical modifications, amphiphilic cellulose achieves suitable properties of biodegradable polymeric surfactants. The design of amphiphilic polysaccharide systems has been sought through the grafting of macromolecules (e.g., hexadecylamine, cholesterol, N-hexadecylacrylamide, acid chloride, acyl chlorides, vinyl laurate or fatty-acid anhydrides) with polymer chains such as carboxymethylcellulose and polylactide or poly(ethylene glycol) (PEG) to chitosan and carboxymethyl chitosan [49].

As to cellulose derivatives, hydrophobically modified O-(hydroxyethyl) cellulose (HEC) is known as a cellulosic surfactant that possesses unique properties in aqueous solution [50]. Depending on the source, structure, and type of cellulose, grafting modification can be performed in different conditions. When the reaction involves native cellulose such as cotton or nanostructured forms (i.e., CNC and CNF), heterogeneous grafting occurs in the presence of ionic liquids with differences as a function of the specific conditions used.

The most commonly produced chemically modified cellulose derivatives are cellulose ethers and esters made by reactions of hydroxyl groups of cellulose. Basing on the way the hydrophobic branch is anchored to the hydrophilic substrate, the graft copolymerization onto cellulose and its derivatives is generally performed by different approaches including [23,51]: “grafting through’’, ‘‘grafting onto’’ and “grafting from’’ as schematized in Figure 5. The “grafting through” technique generally consists of copolymerized premade vinyl-functionalized cellulose with co-monomers. The “grafting onto” technique requires the pre-synthesis of end-functionalized linear chains that are subsequently covalently bonded to the polysaccharides [52]. However, the formation of a high-grafting density is limited by steric hindrance. The “grafting from” technique involves the growth of polymer grafts directly onto a cellulose backbone and is the most used approach due to its attractive advantages: introduction of high-density side-chains (the polymer–monomer reaction is less affected by steric hindrance) and simple purification of final copolymers [53]. The “grafting from” polymerization has been extensively investigated with a particular interest in the development of alternative and tailoring radical polymerization techniques [54]. For example, amphiphilic graft copolymers with a hydrophilic hard polar hydroxypropyl cellulose backbone and hydrophobic soft nonpolar polyisobutylene (PIB) branches have been prepared. The polyisobutylene branch length in the graft copolymers could be designed by living cationic polymerization. The hydrophilicity on the surface of graft copolymer films could be turned hydrophobic by increasing grafting density or length of polyisobutylene branches. (Amphiphilic Graft Copolymers of Hydroxypropyl Cellulose Backbone with Nonpolar Polyisobutylene Branches) [55].

#### 3.1.2. Radical Graft Copolymers

(a)Conventional free-radical polymerization

Free-radical polymerization still represents the most common method of synthesizing polymers in industrial fields due to its versatility [56]. The applicability to the graft copolymerization onto cellulose consists of the addition of free radicals (formed by chemical initiators or irradiations) onto the monomers (i.e., vinyl or acryl) to form a covalent bond on the cellulose backbone and consequently a free-radical site. The latter is the branch involved in the propagation of reaction until termination. Despite the advantages in general setup, reagent purity, wide range of suitable monomers and reaction conditions, the control over graft molecular weight and its distribution is very limited mainly because of some “side” reactions affecting the growing species during polymerization. To overcome this drawback, which is accompanied by the degradation of the cellulose backbone and the collateral formation of homopolymer, the development of alternative radical polymerizations has been tried. The recent development of controlled radical polymerization (CRP) techniques in anionic, cationic, radical, and ring-opening polymerizations has made possible the introduction of an appropriate functional group quantitatively on the terminal of various cellulose polymers. Grafting strategies––including heterogeneous surface modifications (i.e., on fibers or nanofibrils) and homogeneous reaction environments––were applied to the cellulose modification of surface-direct hydrophobic segments anchored on the cellulose backbone.

(b)Controlled radical polymerization

The need to obtain polysaccharide-based hybrids with precise and tailored properties for synthetic graft length, chemical composition and topology, led to development of controlled radical polymerization (CRP). These techniques are tolerant of moisture and compatible with a large range of functional groups. They are based on the addition of species that ensure the reversible trapping of the “active” propagating radical species as “dormant” species through reversible termination or reversible transfer [52]. This strategy makes it possible to reduce the concentration of propagating radical chain ends in order to minimize the occurrence of irreversible termination reactions and thus the formation of “dead” polymer chains.

The recent advent of CRP such as nitroxide-mediated polymerization (NMP), atom transfer radical polymerization (ATRP) or reversible addition-fragmentation chain transfer (RAFT). Among CRP techniques, ATRP is still the most widely used due to its versatility and compatibility with a wide range of vinyl monomers and solvents, and or its solid surface modification [57]. Grafting cellulose via ATRP, a mild approach that does not need any pretreatment with other chemicals, is a providing polymer with controlled molecular weights and low polydispersities. The initiators usually used in ATRP are halogenated compounds with halogen atoms that have been activated by α-carbonyl, phenyl, vinyl or cyano groups [51]. Indeed, the active species generated through a reversible redox process complex successively undergoes one electron oxidation with the concomitant abstraction of the halogen atom, X, from a “dormant” species, R–X [54].

#### 3.1.3. Ring-Opening Polymerization (ROP)

ROP is a well-established technique to polymerize cyclic monomers to prepare aliphatic polyesters of cellulose [58]. The driving force behind it is the release of ring strain, and no treatment of the cellulose is necessary prior to the grafting reaction. Lactones and lactides show excellent properties in terms of biodegradability, biocompatibility, and permeability. In particular, poly(ε-caprolactone) (PCL) and poly(l-lactide) (PLLA), interesting for their applications in biomedical materials, have been widely grafted onto cellulose via ROP [23]. A schematic synthesis procedure is reported in Figure 6. ROP can be efficiently performed on several cellulose forms from solid to crystalline state. Heterogeneous covalent modification with either PCL or PLLA of solid cellulose substrates, e.g., cellulose fibers or fiber networks such as paper [25]. On the other hands, to improve the accessibility of monomers to the cellulose backbone, a highly homogeneous ROP can be conducted on efficiently dissolved cellulose [52]. For this purpose, ionic liquids can be used as suitable green solvents that can disrupt inter-polymer attractions for a high thermal and chemical stability. The ROP of lactones is most commonly performed in presence of metal-based catalyst such as tin-alkoxide species [26]. Particularly suitable is tin 2-ethylhexanoate (Sn(Oct)_2_), the widely used catalyst in the production of biodegradable polyesters due to its low biologic toxicity. Because it avoids transesterification reactions, Sn(Oct)_2_ ensures the synthesis of final graft copolymers in high yield, controlled molecular weight, and narrow distribution. A coordination–insertion mechanism is thought to be initiated by a tin–alkoxide species formed prior to ROP [57].

Cellulose graft copolymers with complex architectures, such as a dual-, block-, or centipede-like graft can be synthesized by combing ROP with other polymerization methods, such as ATRP [57]. For example, combining ATRP and ROP has been used to produce block-copolymer grafts from cellulose derivatives. Berthier et al. grafted HPC to develop a system for fragrance delivery in ethanol-rich solutions. To create copolymer amphiphilic nano-sized particles and micellar structures for drug delivery, the poly(tert-butyl acrylate) block was deprotected to give poly(acrylic acid) by acidolysis and was subsequently cross-linked for preparation [59].

### 3.2. Methods for Cellulose Dissolution

It is well known that cellulose is hard to dissolve because of its morphology and high crystallinity [60]. As previously said, the chemical structure is characterized by a three-dimensional arrangement of hydrogen bonds, connected by hydroxyl groups to each polymer chain. To separate this stable and highly packed structure, an excellent solvent is required to break the intermolecular H-bond interactions, thus favoring the dissolution of the polymer [43].

Several in the last three decades’ methods have been explored to dissolve the cellulose. Traditionally, these have involved defibrillation by viscous processes aimed at inducing the formation of regenerated cellulose fibers. However, these procedures involved the use of aggressive solvents to chemically modify the macromolecules, which caused problems in waste disposal, health safety and environmental pollution [61]. More recently, non-derivative solvents such as N, N-dimethylacetamide/lithium chloride (DMAc/LiCl) or ammonium fluoride/dimethylsulfoxide (DMSO) were successfully used to dissolve cellulose by using N-methyl-morpholine-N-oxide (NMMO) for applications on the industrial scale [62]. Indeed, these solvents present relevant benefits in low toxicity, biotolerability and high recovery rate (>99%), but they also have disadvantages related to the use of high temperatures to dissolve cellulose macromolecules and the occurrence of oxidative side reactions during the process. For this purpose, the use of ionic liquids is a promising alternative strategy for dissolving cellulose at high concentrations (until 20%), thanks to their ability to dissolve cellulose at low temperatures [63]. Taking into account their specific properties, ionic liquids have been successfully used in several fields, including reaction media in organic synthesis or electrolytes for electrochemical applications because they have a lower impact on the environment and human health compared to conventional organic solvents. From a chemical point of view, the ability of ionic liquids to dissolve cellulose basically rises from the presence of anionic groups (i.e., chlorides) working as strong hydrogen acceptors, especially in the presence of microwave heating, that easily solubilize cellulose through hydrogen bonding from hydroxyl functions to the anions [45]. Rogers first reported that ionic liquids efficiently dissolved cellulose, and since then, many of them have been studied for their ability to dissolve cellulose [64]. As a result of its crystalline form, natural cellulose is relatively stable in common solvents [65]. According to transmission electron microscopy studies, cellulose begins to dissolve by destroying its fibers, then gradually dissolves, and finally a mixture of cellulose chains and ionic liquids is formed in a stable state [27]. There have been promising trials to minimize cellulose degradation and investigate new cellulose derivatives and hybrids [66]. and [67]. However, some properties of them including toxicity, poor biodegradability and high processing costs, which drastically limit their use as green solvents [68]. Hence, several groups are recently exploring an green alternative solvent to dissolve cellulose derivatives. Among these, deep eutectic solvents (DESs) are gaining interest for various chemical and biological applications. DESs are made from the combination of a salt and a hydrogen-bond donor (HBD), characterized by a lower melting point compared to those of single components [64] that confer low inflammability due to a low volatility of organic compounds (VOCs). Moreover, they are unreactive with water but dissolve complex biomolecules such as cellulose by a cationic interaction that improved their hydrophilic properties [69]. From this perspective, ionic liquids and DESs currently offer the most promising methods for the sustainable processing of cellulose in industry.

## 4. Applications

### 4.1. Drug Delivery Applications

AC has enormous potential for drug delivery applications due to its high surface area-to-volume ratio and high polymerization [70]. Its high loading and binding capacity to active pharmaceutical ingredients enables it to fine tune the release mechanisms [71,72]. Moreover, the use of amphiphilic copolymers is mandatory for fabricating DESs because the appropriate setting of chemical modification conditions enables them to overcome the intrinsic hydrophobic limits of celluloses. As a function of the molecular grafting strategies, AC may be characterized by different encapsulation efficiency via physical entrapment and tunable pharmaco-kinetic properties, depending upon the local environmental conditions (i.e., temperature, pH). From this perspective, a plethora of drug-delivery systems with different administration strategies involving oral, ocular, intra-tumoral, topical, and transdermal routes have been proposed in recent years [24,28,73,74].

Recently, AC has been successfully used to improve the bioavailability of orally delivered drugs. Several experiments confirmed that the peculiar properties of AC may be modulated to design innovative drug-delivery systems with improved stability in the gastrointestinal tract and poor permeability across the intestinal epithelium to minimize the bioavailability of therapeutic molecules [75]. Accordingly, much effort has been employed to engineer efficient oral drug-delivery systems based on natural polymers with enhanced oral bioavailability (Figure 7) [38,76]. For example, the coating of therapeutic molecules with acid-stable pH polymers improved their stability in the acidic environment of stomach. Recently, microbeads for the controlled release of diclofenac sodium prepared by using carboxymethyl cellulose and chitosan demonstrated a pH-sensitive drug release profile that prevented the initial burst release in the gastrointestinal tract [20,77]. Another efficient strategy is the encapsulation of drugs in a polymeric matrix, such as in ethyl cellulose, which was found to enhance gastrointestinal stability against enzymatic degradation. In addition, to improve oral bioavailability of many poorly water-soluble drugs, lipids, surfactants and prodrugs, amphiphilic polymers have been investigated as a promising candidate in this field, and amphiphilic cellulose-based materials exhibit a potential means to improve oral drug bioavailability [22,78,79].

Carboxylated cellulose nanocrystals have gained a promising interest due to the presence of carboxylic acid groups. Wang et al., prepared a series of amphiphilic carboxylated cellulose-graft-poly(l-lactide) copolymers via the ROP technique. The solubility of the graft copolymers in organic solvents improved. The prepared amphiphilic copolymers were self-assembled into nanoparticles for delivery of anticancer drug oleanolic acid. The copolymer nanoparticles displayed a high drug-loading efficiency and a prolonged drug release. The amphiphilic carboxylated cellulose-graft-poly(l-lactide) copolymer nanoparticles provided a novel platform for drug-delivery applications (Figure 8) [22]. In the field of amphiphilic cellulose preparation through the grafting process, Li et al., prepared an ethyl cellulose graft of amphiphilic poly(ethylene glycol) methyl ether methacrylate via atom transfer radical polymerization (ATRP). The self-assembly and thermosensitive property of the obtained ethyl cellulose-g-P(PEGMA) amphiphilic copolymers exhibited the formation of spherical micelles in water with a lower critical solution temperature of around 65 °C [80].

In a different study, amphiphilic cellulose such as carboxymethyl cellulose acetate and carboxymethyl cellulose acetate butyrate were synthesized using cellulose pup, extracted from bagasse. These two amphiphilic cellulose derivatives were applied as stabilizers for hydrophobic drugs like sulfadiazine in a water dispersion. Carboxymethyl cellulose acetate butyrate exhibited a more efficient drug loading capacity (42.88%) than carboxymethyl cellulose acetate (1.78%) which may have been related to the high degree of substitution of the hydrophobic parts [81].

It was reported that up to 80% of drugs suffer from poor bioavailability due to poor aqueous solubility. Various methods have been established to improve drug solubility and bioavailability. Amorphous solid dispersions represent some of the most efficient formulations to improve drug solution concentration. The polymers prepared for amorphous solid dispersions matrices have shown certain structural criteria: (1) a terminal carboxyl group interacts with the drug and can act as a pH trigger to enhance the release process; (2) a degree of hydrophobicity controls miscibility with hydrophobic drugs; (3) a degree of hydrophilicity controls the release of a drug in an aqueous environment, and (4) a sufficiently high glass-transition temperature (Tg) that can immobilize the drug, prevents recrystallization and ensures that polymer dispersion remains in the glassy state [41].

Cellulose ω-carboxyesters have been established as effective matrix polymers for amorphous solid dispersion applications with various drug molecules. The moderate hydrophilicity arising from carboxylic acid groups promotes its amphiphilicity to work effectively as amorphous solid dispersion matrices [82]. Dong et. al., reported that the tandem cross-metathesis/thiol Michael procedure, followed by saponification where appropriate, enables multifunctional modification of cellulose ether derivatives. This technique permits the synthesis of different carboxyl-containing polysaccharides specifically aimed for amorphous solid dispersion [83].

Also, olefin cross-metathesis was reported as z promising technique for preparing amphiphilic derivatives of hydroxypropyl cellulose. The preparation of olefin-terminated hydroxypropyl cellulose derivatives followed by cross-metathesis with various acrylates and hydrogenation afforded stable, saturated products. The 5-carboxypentyl hydroxypropyl cellulose derivative showed high promise as a crystallization inhibitor of telaprevir from a supersaturated solution. Previous articles showed that amphiphilic cellulose has substantial promise in drug-delivery and amorphous solid-dispersion applications [41].

Zhenzhen Liu et al., reported that cationic amphiphilic cellulose copolymers could be prepared through grafting hydrophobic poly (p-dioxanone) chains onto hydrophilic quaternized cellulose derivatives via a ring-opening polymerization reaction, which was performed in 1-butyl-3-methylimidazolium chloride using 4-dimethylaminopyridine or 1,8-diazabicyclo (5.4.0) undec-7-ene (DBU) as s catalyst, (Figure 9). Studying the self-assembly of the grafted cellulose showed that the size and critical micelle concentration of the formed micelles decreased with increasing grafting content of poly (p-dioxanone) chains. The ζ-potentials of the micelles were cationic and ranged from 39.1 to 45.4 mV. The highest encapsulation efficiency of paclitaxel (PTX) into the micelles was 61.8%, and 92.0% of the loaded PTX was continuously released from the micelles [45].

Thermo-responsive micelles were prepared by reductive amination between hydroxypropyl methyl cellulose containing an amine group (monoamine, diamine, or triamine JEFFAMINE) as hydrophobic block. The reaction produced diblock, triblock and three-armed copolymers with different hydrophilic/hydrophobic ratios. The geometrical structure of copolymers strongly affected the micelle size as well as the cloud point of the hydroxypropyl methyl cellulose-JEF copolymers. Spherical nano-micelles were formed by the self-assembly of copolymers in aqueous solution, and the micelle size was tailored by varying the block length of the HPMC and the geometrical structure. Three-armed HPMC–-JEF copolymers presented a lower critical micelle concentration and smaller micelle size compared to linear diblock and triblock ones. MTT present outstanding cytocompatibility, suggesting that these novel HPMC-JEF copolymers can be safely used as a potential drug carrier [84].

### 4.2. Other Applications

#### 4.2.1. Immobilization

Various forms of amphiphilic cellulose were prepared and characterized for different industrial applications. The most promising approach was the use of amphiphilic cellulose materials to develop support for immobilizing nanoparticles. For example, Li et al., prepared cellulose sponges through dual-cross-linking cellulose nanofibers with γ-glycidoxypropyltrimethoxysilane and polydopamine. The highly porous and flexible cellulose sponge was used as a carrier for palladium nanoparticles, which are homogeneously dispersed on the surface of a cellulose sponge with a narrow size distribution. The Pd@cellulose sponge exhibited high catalytic activity and efficient recyclability in Suzuki and Heck cross-coupling reactions [85].

Amphiphilic cellulose was synthesized by grafting hydrophobic groups onto hydrophilic cellulose nanocrystals and used as a support for PdNi alloy nanoparticles. The as-synthesized catalysts exhibited encouragingly high performance in the hydrodeoxygenation of vanillin under mild conditions [86]. Amphiphilic cellulose was fabricated by introducing a hydrophobic cetyl group in hydrophilic CNCs. PdNi bimetallic nanoparticle catalysts were prepared by depositing PdNi-alloy NPs directly onto the modified CNCs and tested in the hydrodeoxygenation reaction of vanillin [85].

Tajima et al., investigated nanofibrillated bacterial cellulose produced by culturing a cellulose-producing bacterium in a medium supplemented with carboxymethylcellulose will aggregate in organic solvents, -i.e., a cellulose-producing bacterium named Gluconacetobacter intermedius NEDO-01 in a medium supplemented with hydroxypropylcellulose. The prepared hydroxypropyl-nanofibrillated bacterial cellulose was dispersible in various polar organic solvents, and this property has valuable applications in molding and complexation [87].

#### 4.2.2. Emulsion Stabilization

Recent trials have been carried out to understand the stabilization mechanism as a function of the amphiphilic properties of the nanocrystals through tuning the surface-charge density. Kalashnikova et al., used two sources of cellulose: cotton linters and bacterial cellulose to explore their emulsifying capability. The results showed that the electrostatic interaction plays the main role in the control of the interface. It has been demonstrated that a decrease in the charge density of cellulose nanocrystals below 0.03 e/nm^2^ efficiently stabilized the oil/water interface [88].

Another strategy for preparing amphiphilic CNC is by chemical pretreatment of cellulose with periodate oxidation followed by reductive amination. This strategy may suggest the preparation of cellulose nanofibers without the widely used acid hydrolysis. Visanko et al., used three butylamine isomers to fabricate CNCs with amphiphilic features. The characterization tools showed that iso- and n-butylamine attach the highest number of butylamino groups to the cellulose fibers [89]. The presence of these alkyl groups increased the hydrophobic nature of the CNCs. The hydrophobic character originated from the presence of butylamino groups on the backbone. Of the three butylamine isomers, iso- and n-butylamine reacted the most efficiently with the dialdehyde groups, whereas tert-butylamine had the lowest conversion rate to butylamino groups. The results proved that isobutylamino-functionalized CNCs were the most hydrophobic (110.5°) and the tert-butylamino-functionalized CNCs were the most hydrophilic (66.9°). The amphiphilic nature of the butylamino-functionalized CNCs was used to stabilize the oil/water emulsions where the iso- and n-butylamino groups were the most effective to forestall the coalescence of the oil droplets.

#### 4.2.3. Adsorption

Alternative approaches involve the fabrication of nano-fibrillar aerogels by coupled TEMPO oxidation and mechanical blending assemble into monolithic forms of porous structures [90]. In this case, amphiphilic aerogels were subjected to vapor deposition with triethoxyl(octyl) silane which turned the amphiphilic aerogel more hydrophobic and oleophilic, thereby showing optimal features to be used as super-absorbents for selective oil removal and recovery. Carboxymethyl cellulose acetate butyrate has application in the waterborne coating industry because it makes the coating application more consistent and defect free. It also improves flow and leveling combined with sag resistance and reduces drying time. The preparation and the properties of carboxymethyl cellulose acetate butyrate as an amphiphilic polymer were reviewed [46].

#### 4.2.4. Membranes and Bacterial Growth Inhibition

Gao et al., prepared amphiphilic cellulose in cellulose NaOH/urea aqueous solution. Amphiphilic cellulose was blended with polysulfone as an additive for preparing porous membranes. The hydrophobization chain segment of amphiphilic cellulose provided compatibility between cellulose and polysulfone, and hydrophilic, and antifouling protection were then created from the hydroxyl surface of amphiphilic cellulose. The water flux and the tensile strength were enhanced after adding nanosized amphiphilic cellulose into the membrane [91].

Films prepared from cellulose nanofibrils are commonly brittle, which may limit its application. Various additives have been proposed to increase the ductility of nanocellulose films [92]. Among these, the impact of polyethylene glycol on both flexibility and film strength was investigated, as was its effect on the morphology and physical properties of nanocellulose films.

Surface curvatures and surface anti-wetting synergistically act to inhibit bacterial contamination. Several trials were carried to transform the hydrophilic character of cellulose material into an amphiphilic character. Jin et al. prepared titania–PFOTMS composite thin films onto the cellulose nanofiber by chemical etching and coating [93]. The prepared composite cellulose sheet exhibited excellent anti-wetting effects and inhibition effects to *E. coli*. The results showed that amphiphilic cellulose materials are promising for inhibition the growth of bacteria.

## 5. Conclusions

Amphiphilic cellulose materials are composed of hydrophobic and hydrophilic moieties; the former ones enable polymer solubility in aqueous solvents, while the latter ones enable the formation of polymeric micelles in aqueous solvents due to hydrophobic interactions. These peculiar features in combination with the high flexibility of these materials to be processed allows fabricating cellulose in different forms (i.e., films, fibers, hydrogels, and aerogels) with an enormous potential in different applications in different areas, i.e., biomedicine, drug delivery, bio-environment. 

## Figures and Tables

**Figure 1 pharmaceutics-14-00386-f001:**
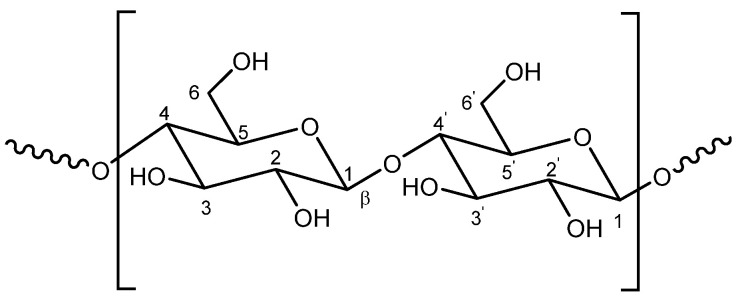
Schematic representation of the structure of cellulose.

**Figure 2 pharmaceutics-14-00386-f002:**
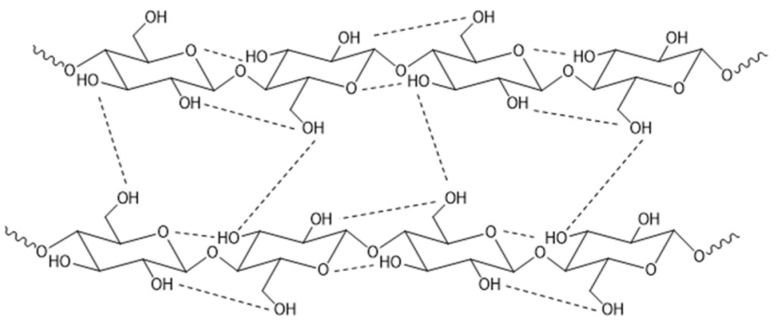
Structure and hydrogen bonds pattern in cellulose: intra- and inter-chain interactions.

**Figure 3 pharmaceutics-14-00386-f003:**
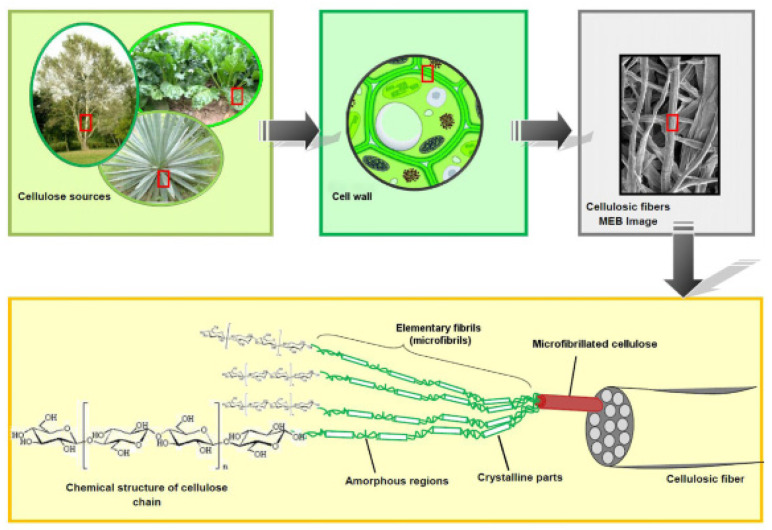
Schematic description of cellulosic fiber structure adapted with the permission from Elsevier, 2012 [5].

**Figure 4 pharmaceutics-14-00386-f004:**
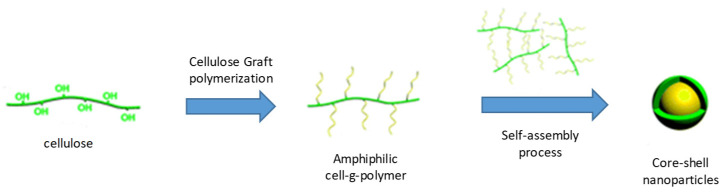
Schematic representation of cellulose grafting polymerization and self-assembly of amphiphilic cellulose graft polymers in core-shell structures.

**Figure 5 pharmaceutics-14-00386-f005:**
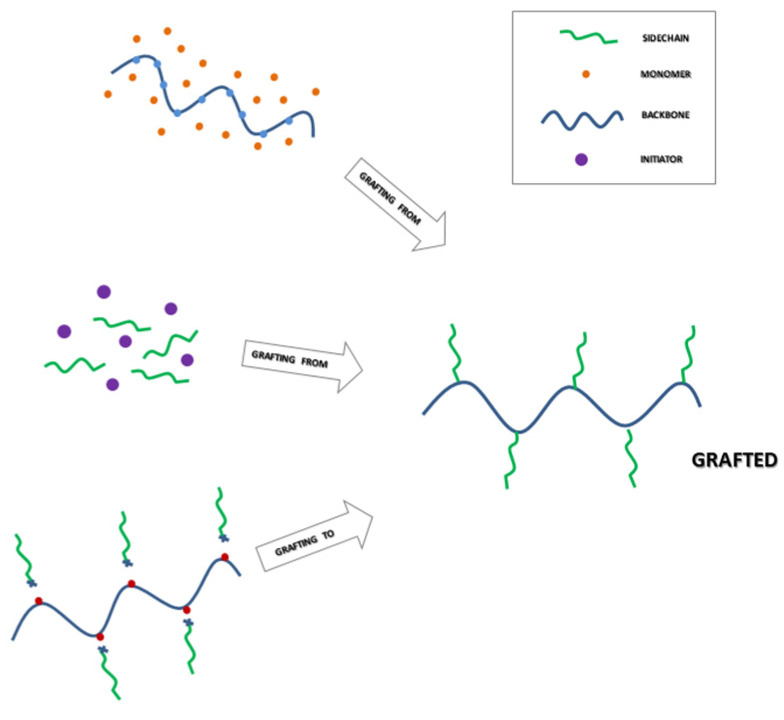
Grafting strategies for the preparation of grafted co-copolymers. The image is freely inspired from [53].

**Figure 6 pharmaceutics-14-00386-f006:**
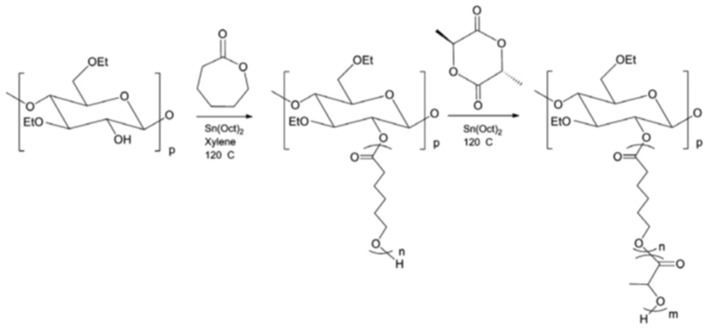
Scheme of cellulose graft copolymers via a homogeneous ROP.

**Figure 7 pharmaceutics-14-00386-f007:**
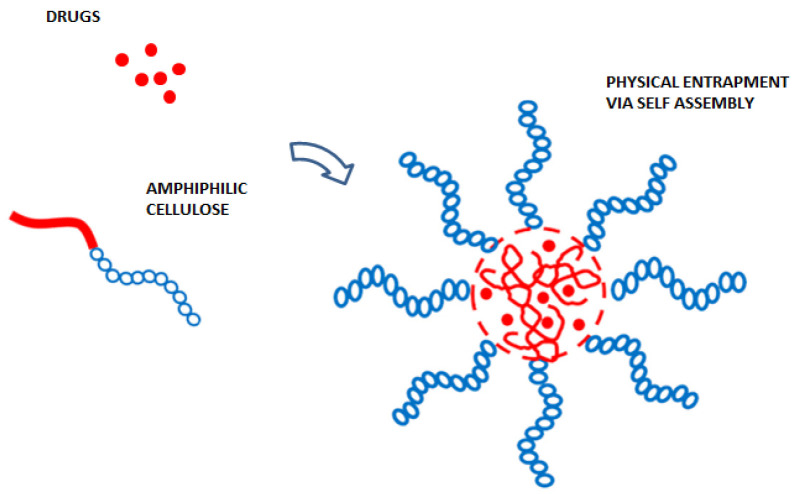
Scheme of the self-assembly mechanism of amphiphilic cellulose for drug delivery.

**Figure 8 pharmaceutics-14-00386-f008:**
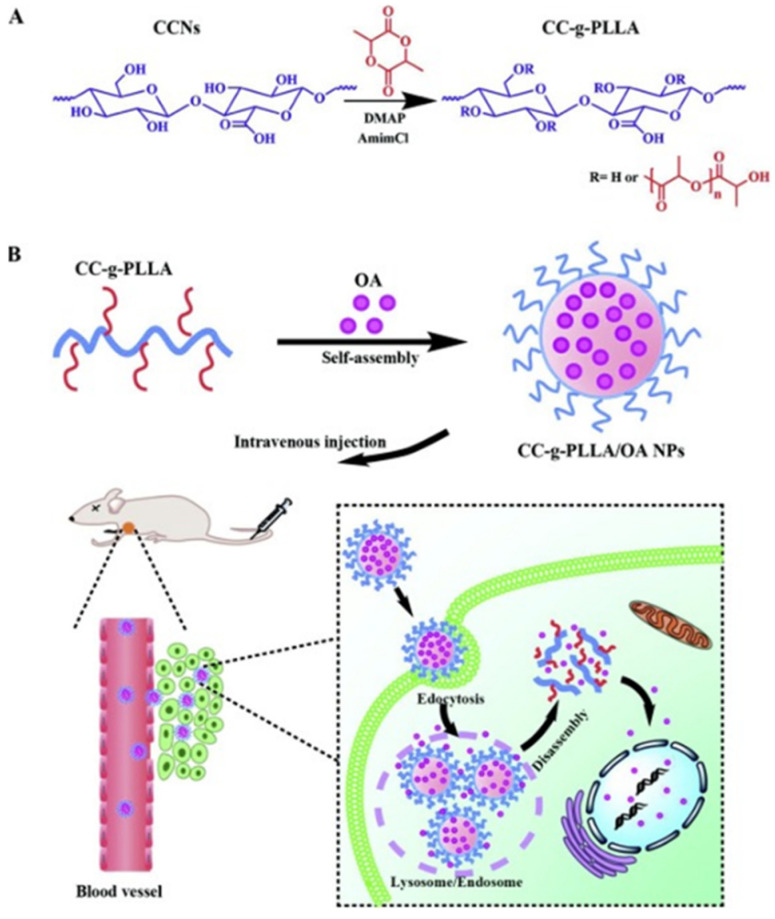
(**A**) Carboxylated cellulose-g-poly(l-lactide) copolymers to improve the drug solubility. In the square, (**B**) the description of release mechanisms for cell inhibition to enhance antitumor efficiency. Adapted with the permission of Elsevier, 2019 [22].

**Figure 9 pharmaceutics-14-00386-f009:**
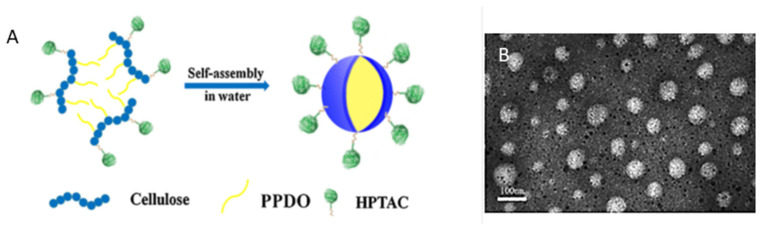
Nanosized quaternized cellulose-g-poly (p-dioxanone) copolymers in aqueous water: (**A**) Scheme of preparation and (**B**) TEM image. Adapted with permission from Elsevier, 2019 [45].

**Table 1 pharmaceutics-14-00386-t001:** Summary of amphiphilic celluloses as a function of Processing, source, main properties and applications.

Process	Cellulose Source	Properties	Applications	References
Grafting	Carboxymethyl cellulose	Self-assembled micellar nanoparticles	Drug delivery system	[19]
	Hydroxyl propyl methyl cellulose	Thermo-responsive micelles		[20]
	Cellulose nanocrystals	Amphiphilic catalyst support	Biofuel upgrade under mild conditions	[21]
	Bacterial cellulose	Nanofibrillated cellulose	Filler material	[22]
Controlled graft polymerization	Cellulose fibers	Comb-shaped amphiphilic cellulose	Surfactants, rheology modifiers, drug carriers, surface modifiers, and polymer blend compatibilizers	[23]
Atom transfer radical polymerization	Ethyl cellulose	Self-assembled spherical micelles	Drug delivery	[24]
Ring opening polymerization	Cellulose fibers	Cellulose-g-PCL Cellulose-g-PLLA	Surface modification Drug delivery systems Biomedicine	[25,26,27]
	Carboxylated cellulose Nanocrystals Bacterial cellulose	Nanoparticles		[28]

## Data Availability

Not applicable.

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
