# Peer review of "Cellulose Amphiphilic Materials: Chemistry, Process and Applications"

_pharmaceutics, 2022, doi:10.3390/pharmaceutics14020386_

Round 1

Reviewer 1 Report

In this review, authors introduced cellulose and different cellulose forms first, and then introduced the properties, preparation methods and practical applications of amphiphilic cellulose. However, the authors need to provide more literature to support the thesis. Besides, this paper still has some problems in format and content. Some comments are listed as followed:

  1. Page2, L46~L51, when introducing the four cellulose crystal types, the authors can add some characteristics about the crystal types, such as the difference of the XRD characteristics of the four crystal types.
  2. Page2, L50, the citation format is incorrect: “mercerization4”.
  3. Page4, L139, the citation format is incorrect: “micrometers14”.
  4. Page4, L122, the “CCNs” abbreviation is wrong.
  5. Page 6, L221, the abbreviation “AC” is inconsistent with the previous description.
  6. Authors need to provide high resolution of figure 6.
  7. Page14, L618, a spelling mistake: “itt” should be changed to “it”.
  8. Page14, L614~L635, describes the method of CNC synthesis of amphiphilic cellulose and should not be placed here.
  9. The part4: processing routes, has no obvious relevance to the amphiphilic cellulose. Can it be deleted?
  10. When introducing various cellulose materials (e.g., cellulose nanocrystals, cellulose nanofibers, bacterial cellulose and cellulose derivatives), authors should introduce examples of how each cellulose material can be used to synthesize amphiphilic cellulose with illustrations rather than simple introductions.
  11. When introducing the synthetic method of amphiphilic cellulose, authors should cite and illustrate examples of amphiphilic cellulose synthesized using this method in the literature, rather than just introducing the method.
  12. Not only immobilization and adsorption are listed in 5.3, but also the bacteriostatic effect of amphiphilic cellulose is described in page15, L658~L664. Therefore, it is suggested that the authors describe each application separately and add examples, preferably graphics with examples, to elaborate.
  13. The authors can make a table comparing the sources, synthesis methods, properties and applications of different amphiphilic cellulose.
  14. Each figure in this review is too thin to elaborate on the problem, there are too few citations.
  15. Copyright permission is required for citation figures.
  16. Authors should standardize the format of references.
  17. The authors need to supplement the conclusions and outlook of the review.

Author Response

In this review, authors introduced cellulose and different cellulose forms first, and then introduced the properties, preparation methods and practical applications of amphiphilic cellulose. However, the authors need to provide more literature to support the thesis. Besides, this paper still has some problems in format and content. Some comments are listed as followed:

Page2, L46~L51, when introducing the four cellulose crystal types, the authors can add some characteristics about the crystal types, such as the difference of the XRD characteristics of the four crystal types.

Thank you for the comment, the text has been amended by including a sentence that underline the crystal types (see yellow marked text at page 2).

Page2, L50, the citation format is incorrect: “mercerization4”.

Thank you for the comment, the text has been amended.

Page4, L139, the citation format is incorrect: “micrometers14”.

Thank you for the comment, the text has been amended.

Page4, L122, the “CCNs” abbreviation is wrong.

Thank you for the comment, the text has been amended.

Page 6, L221, the abbreviation “AC” is inconsistent with the previous description.

Thank you for the comment, the text has been amended

Authors need to provide high resolution of figure 6.

Thank you for the comment, the figure has been improved as requested.

Page14, L618, a spelling mistake: “itt” should be changed to “it”.

Thank you for the comment, the text has been amended

Page14, L614~L635, describes the method of CNC synthesis of amphiphilic cellulose and should not be placed here.

Thank you for the comment, these sentences have been moved in the paragraph 2.1 Cellulose Nanocrystals (CNCs), at page 19.

The part4: processing routes, has no obvious relevance to the amphiphilic cellulose. Can it be deleted?

Thank you for the comment, The paragraph 4 has been removed as requested. Subparagraph 4.1 now is indicated as 3.2. Moreover, the paragraph 4.2 has been removed as suggested.

When introducing various cellulose materials (e.g., cellulose nanocrystals, cellulose nanofibers, bacterial cellulose and cellulose derivatives), authors should introduce examples of how each cellulose material can be used to synthesize amphiphilic cellulose with illustrations rather than simple introductions.

Thank you for the comment, the text has been amended by including different examples as requested (see yellow marked text at page 4 and 5 ).

When introducing the synthetic method of amphiphilic cellulose, authors should cite and illustrate examples of amphiphilic cellulose synthesized using this method in the literature, rather than just introducing the method.

Thank you for the comment, the text has been amended by including some examples as requested (see yellow marked text at page 9).

Not only immobilization and adsorption are listed in 5.3, but also the bacteriostatic effect of amphiphilic cellulose is described in page15, L658~L664. Therefore, it is suggested that the authors describe each application separately and add examples, preferably graphics with examples, to elaborate.

Thank you for your comment. The paragraph 5.3 (now 4.2) has been divided in 4 subparagraphs according to the reviewer suggestion

The authors can make a table comparing the sources, synthesis methods, properties and applications of different amphiphilic cellulose.

Thank you for your comment. A table has been included according to the reviewer suggestions

Each figure in this review is too thin to elaborate on the problem, there are too few citations.

Thank you for your comment. Several new references have been included in different part of the manuscript, also in the application section, in agreement with your suggestion (see yellow marked refs).

Copyright permission is required for citation figures.

Thank you for the comment. All permissions for figures have been required at the moment of the first submission.

Authors should standardize the format of references.

Thank you for the comment. References layout has been checked.

Reviewer 2 Report

The manuscript (pharmaceutics-1546014-peer-review-v1) reviewed the customization of different amphiphilic cellulose materials from various resources such as cellulose nanofibers, bacterial cellulose and cellulose derivatives, And the application prospect of amphiphilic cellulose in the fields of drug delivery, molecular / particle adsorption and so on. About this manuscript, the reviewer has the following questions:

  1. Adjust and unify the layout and format of the manuscript.
  2. Please increase the proportion of references published in recent years.
  3. Explain the Hansen solubility parameters mentioned in the manuscript. What is their significance?
  4. Ionic liquids have been applied as solvent and reaction media for cellulose reactions, so how did it improve the solubility of cellulose? What are the advantages over other solvents?
  5. As a drug delivery material, what is the mechanism of AC loading drugs? Whether it has good biocompatibility in vivo?
  6. It is suggested that the discussion on adsorption should be added in Section 5.2.
  7. Related cellulose application references in biomedical fields are suggested to cite, such as: 1) 10.1016/j.ijbiomac.2019.01.191; 2) 0.1016/j.ijbiomac.2020.10.046; 3) 10.1016/j.ijbiomac.2019.01.140.

Author Response

The manuscript (pharmaceutics-1546014-peer-review-v1) reviewed the customization of different amphiphilic cellulose materials from various resources such as cellulose nanofibers, bacterial cellulose and cellulose derivatives, And the application prospect of amphiphilic cellulose in the fields of drug delivery, molecular / particle adsorption and so on. About this manuscript, the reviewer has the following questions:

Adjust and unify the layout and format of the manuscript.

Thank you for the comment. References layout has been checked.

Please increase the proportion of references published in recent years.

Thank you for your comment. Some newest references have been included in the applications section (see from 60 to 67)

Explain the Hansen solubility parameters mentioned in the manuscript. What is their significance?

Thank you for the comment. The Hansen parameters identify a theoretical model to study the solubility of liquid solution. A new reference has been cited to support the significance for readers. [Novo L.P., Curvelo A.A..S. Hansen Solubility Parameters: A Tool for Solvent Selection for Organosolv Delignification. Ind. Eng. Chem. Res. 2019, 58, 31, 14520–14527]

Ionic liquids have been applied as solvent and reaction media for cellulose reactions, so how did it improve the solubility of cellulose? What are the advantages over other solvents?

Thank you for the comment, the text has been amended by addind some comments on cellulose dissolution in ionic liquids (see yellow marked text at page 11)

As a drug delivery material, what is the mechanism of AC loading drugs? Whether it has good biocompatibility in vivo?

Thank you for your comment. Some sentences have been included in the paragraph 4.1 on drug delivery systems according with the reviewer suggestions, as follows: “Besides, the use of amphiphilic copolymers is mandatory to fabricate DDS because enables to overcome intrinsic limits of celluloses in terms of hydrophobicity, by an appropriate setting of chemical modification conditions for their functionalization. As a function of the molecular grafting strategies, AC may be characterized by different encapsulation efficiency - via physical entrapment - and tunable pharmaco-kinetic properties, depending upon the local environmental conditions (i.e., temperature, pH). In this view, in the last years, a plethora of drug delivery systems with different administrations strategies that involve oral, ocular, intra-tumoral, topical, and transdermal routes have been proposed [63-66]”.

It is suggested that the discussion on adsorption should be added in Section 5.2.

Thank you for this comment. the discussion on adsorption should be added in Section 4.2

Related cellulose application references in biomedical fields are suggested to cite, such as: 1) 10.1016/j.ijbiomac.2019.01.191; 2) 0.1016/j.ijbiomac.2020.10.046; 3) 10.1016/j.ijbiomac.2019.01.140.

Thank you for the comment, references have been included Refs [70-72]

Reviewer 3 Report

The paper concerns amphiphilic celluloses especially from the point of view of the chemistry, preparation and properties.The review is clearly organized and understandable.

My only observation is that, considering the pharmaceutical target of the Journal, some more information, if available, about the pharmaceutical applications of the described derivatives could make the review more interesting for the readers 

Author Response

The paper concerns amphiphilic celluloses especially from the point of view of the chemistry, preparation and properties. The review is clearly organized and understandable.

My only observation is that, considering the pharmaceutical target of the Journal, some more information, if available, about the pharmaceutical applications of the described derivatives could make the review more interesting for the readers.

Thank you for this comment. The paragraph 5.2 (now 4.2) has been revised according with the reviewer  suggestion.